# Assessment of Gus Expression Induced by Anti-Sense *Os*PPO Gene Promoter and Antioxidant Enzymatic Assays in Response to Drought and Heavy Metal Stress in Transgenic *Arabidopsis thaliana*

Zakir Ullah [1], Javed Iqbal [2,*], Banzeer Ahsan Abbasi [3], Wasim Akhtar [4], Sobia Kanwal [5], Iftikhar Ali [6,7], Wadie Chalgham [8], Mohamed A. El-Sheikh [9] and Tariq Mahmood [1,*]

1. Department of Plant Sciences, Faculty of Biological Sciences, Quaid-i-Azam University, Islamabad 45320, Pakistan; zakirullah@bs.qau.edu.pk
2. Department of Botany, Bacha Khan University, Charsadda 24420, Pakistan
3. Department of Botany, Rawalpindi Women University, 6th Road, Satellite Town, Rawalpindi 46300, Pakistan; benazirahsanabbasi786@gmail.com
4. Department of Botany, University of Azad Jammu and Kashmir Muzaffarabad, Muzaffarabad 13100, Pakistan; wasimakhtarqau@gmail.com
5. Department of Biology and Environmental Sciences, Allama Iqbal Open University, Islamabad 44310, Pakistan; sobia.kanwal@aiou.edu.pk
6. Center for Plant Sciences and Biodiversity, University of Swat, Charbagh 19120, Pakistan; iftikharali@uswat.edu.pk
7. School of Life Sciences & Center of Novel Biomaterials, The Chinese University of Hong Kong, Hong Kong 999077, China
8. Department of Mechanical and Aerospace Engineering, University of California, Los Angeles, CA 90095, USA
9. Botany and Microbiology Department, College of Science, King Saud University, Riyadh I1451, Saudi Arabia; melsheikh@ksu.edu.sa
* Correspondence: javed89qau@gmail.com (J.I.); tmahmood@qau.edu.pk (T.M.)

**Abstract:** Abiotic stresses, including drought and heavy metals, are detrimental to plant growth and development and enormously reduce agricultural yields. Plants may quickly change their transcriptome in response to various stressful conditions. Plants develop many defense mechanisms to respond to various stresses that can be classified into morphological, physiological, and biochemical responses. Polyphenol oxidases (PPOs) are one of the self-protective enzymes found in plants except for Arabidopsis. Currently, drought and heavy metals were applied exogenously to transgenic *A. thaliana* lines (transformed with *Oryza sativa* PPO promoter fused to the GUS reporter gene). The current study mainly focused on the systematic pathway by which plants respond to stressors. The aim of this study is to investigate the effect/expression of PPO and antioxidant defense system against abiotic stresses. *A. thaliana* was treated with different concentrations of polyethylene glycols. At 30% PEG, maximum fold induction (1.9) was seen after 12 h. Overall, various concentrations (5%, 20%, and 30%) induced PPO expression after 6, 12, and 24 h. Moreover, three different concentrations of Cu (50 μM, 100 μM, 200 μM) and Ni (50 μM, 100 μM, 200 μM) for 6, 12, and 24 h were also applied. It was observed that the expression profiling of the *Os*PPO promoter induced GUS gene expression in response to Cu and Ni treatments. The maximum fold induction (15.03) of GUS was observed in 100μM of Cu after 24 h. In the case of Ni, maximum fold induction of (7.78) was observed at 100 μM after 24 h. So, both Cu and Ni showed a similar pattern of induction at 100 μM after 24 h. In conclusion, the efficiency of the PPOGUS promoter can be operated to assess the response of plants to various abiotic stimuli.

**Keywords:** *Arabidopsis thaliana*; *Oryza sativa*; transgenic; PPO; GUS; Cu; Ni; drought; APX; CAT; POD

## 1. Introduction

Plants are sessile beings; they are subjected to a wide range of stresses, including drought, salinity, UVs, and heavy metal (HM) stresses [1,2]. Worldwide food security is being challenged by fast population growth and abrupt climate change [3]. Abiotic factors (such as drought, heavy metals, salt, etc.) have a significant impact on crop production [1]. Drought, heat, and heavy metal stresses have become the major limiting factors to agricultural growth and, eventually, food security threats due to climatic change [4]. As one of the most important abiotic stresses, drought limits plant growth, development, and productivity. Drought is becoming more common across the world as precipitation levels decline and rainfall patterns shift [5]. Drought is a severe danger to the viability of food production systems around the globe due to rapidly changing climate dynamics [6]. For instance, by 2050, spring-sown crops in southern Europe are predicted to have wide-spread losses in crop output (e.g., legumes −31 to +5.5%, sunflower −11 to +3.5%, and tuber crops) and increases in water requirement (e.g., maize +3 to +5%, potato +7 to +11%) [7]. The effects of drought stress on plant growth and output are significant [8]. The shortage of moisture and lack of water causes the seeds to suffer in arid regions, lowering their germination probability. Early on in water scarcity, plant cells are affected because the root system uses less water, and the leaves lose water through transpiration. Plant growth is influenced by the osmotic and ionic equilibrium [9]. The methods by which plants respond to drought, comprising morphological traits, biochemistry, and molecular regulation, have been thoroughly studied by scientists to adapt to the arid environment [1,2]. Plants are classified into three categories for resisting droughts: drought-avoidant, drought-tolerant, and drought-resistant varieties. To prevent water loss through transpiration, plants typically modify the structure of their roots and stomata, as well as their cellular metabolism, to better absorb water from the soil [10].

Plants primarily use their root systems to detect water level changes, and the root system's length and surface area affect how well soil water may be absorbed. Severe scarcity reduces food production significantly by interfering with plant development, physiology, and reproduction [11,12]. The intensity of the drought's effects is typically unexpected. It is influenced by a variety of factors, including rainfall patterns, soil holding capacity, and water losses through evapotranspiration. Drought disturbs growth, nutrient and water interactions, photosynthesis, and assimilation partitioning, resulting in a considerable drop in agricultural production [13]. Plant responses to drought stress vary by species and rely on the plant development stage and other environmental conditions [14]. Due to inadequate soil moisture, the key yield-producing elements are reduced. Similarly, absorption of photosynthetically active components, radiation use efficiency, and harvest index retard [15]. Plants adjust their growth patterns and physiological processes in response to the severe impacts of drought stress [16]. Drought stress could, at the cellular level, induce an excessive formation of reactive oxygen species (ROS), which affect cell homeostasis, resulting in oxidative stress and damage to plants in ways that primarily manifest as a decline in photosynthetic efficiency, peroxidation-related cell damage, and cell membrane stability [17]. Enzymatic antioxidants could scavenge ROS, decrease crop damage, and increase stress tolerance. Some common enzymatic antioxidants consist of peroxidase (POD), superoxide dismutase (SOD), catalase (CAT), guaiacol peroxidase (GPX), and ascorbate peroxidase (APX) [18].

Plants rely profoundly on soil solutions to obtain nutrients for their growth and developmental cycle. One of the main causes of crop production loss is the current rise in heavy metal (HM) pollution of arable areas [19]. Large-scale exposure to heavy metal contamination puts the viability of agricultural and environmental systems at risk. Due to inefficient irrigation techniques, excessive use of chemical fertilizers, and other synthetic nutrients, crops are frequently exposed to metal poisoning [20]. HMs are considered to hamper a protein's biological activity by altering its natural shape by binding to it [21]. Like drought, heavy metal ions cause oxidative stress at the cellular level by producing reactive oxygen species (ROS) [22]. They damage DNA and block DNA repair pathways,

alter the functional integrity of membranes and nutritional homeostasis, and affect protein function and activity [23]. Plants have a variety of mechanisms for coping with these challenges. Stress-inducible promoters control stress-related genes, which can be detected in transgenic model plants under stress [24]. Stress-inducible promoters play a pivotal role in the traumatic agent response [25]. Plants have been researched for abiotic stress-inducible promoters to better understand stress response mechanisms [26]. These promoters include cis-regulatory regions that bind stress-responsive factors, causing downstream expression to be activated. A phytoene synthase (DcPSY) promoter from carrots was recently discovered to be triggered by ABA and salt [27].

Likewise, polyphenol oxidases found in plants are highly diverse. They are ubiquitous and are copper-containing, nuclear-encoded enzymes [28]. Various potential compounds such as (flavonol, anthocyanin, flavone, iso-flavonoid, hydroxybenzoic acid, and hydroxycinnamic are substrates for PPO [29]. PPOs are essential in plant defense against biotic and abiotic stressors and can suppress the activity of pathogens by quinone product which is the antimicrobial activity of PPO. This quinone produces an oxidative form of oxygen [30]. An important factor of PPO in plants is the production of special metabolites, but this issue is now unexplored but still a vital compound against an external agent [31]. The phenolic content of osmotically stressed rice seedlings was found to be much higher than that of well-watered seedlings. Under osmotic stress, total phenolics were accumulated in larger quantities (93%) in a shoot [32]. The increased amount of phenolics in the shoot, along with decreased PPO activity, was made to withstand osmotic stress [33]. Furthermore, ultraviolet (UV) and salt (oxidative stressors) were found to promote PPO activity in plants [34]. *Trigonella foenum-graecum* calli grown on a substrate treated with sodium chloride (NaCl) displayed greater PPO activity. Meta QTL (MQTL) analysis identified genomic regions related to ionomic traits in *Arabidopsis thaliana*. It defines reliable genetic markers for breeding ionomic traits via marker-assisted selection. MQTL also indicates the physiological and genetic relationship between these traits [35]. On the physiology of PPOs, there are still many unanswered concerns. Although it is generally accepted that PPOs contribute to biotic and abiotic stress defense responses, the precise method by which they carry out this function is still up for debate. According to [36], they may act through the direct poisonousness of quinones, lowered bioavailability, and alkylation of cellular proteins to pathogens, drought, and heavy metal stressors [37]. Physical barriers are created by the crosslinking of quinones with proteins or other phenolics, as well as the generation of ROS, which is known to be crucial for defense signaling. Agronomically, four significant genes, namely *cry*IA105, *cry*IIIAa, CP4-*epsps*, and *gox*, were engineered in sugar beets to withstand biotic stresses, and improved regeneration of sugar beet plantlets was observed. Agrobacterium tumefaciens LBA4404 and GV3101 strains formed using recombinant plasmids were found to be efficient and suggested inserting new genes in sugar beets [38]. The use of the PPO expression system for the development of stress-tolerant plant species, particularly in crops that may experience harsh simultaneous environmental challenges such as drought, salt, and diseases, appears to be a promising tool for the future. Indeed, numerous transgenic plants have proven effective through PPO transformation, offering tolerance to diverse abiotic and biotic stressors. The goal of this study is to investigate the effect/expression of PPO and the antioxidant defense system against abiotic stresses.

## 2. Materials and Methods

### 2.1. Construct Preparation

The *Os*PPO promoter was found in NCBI, amplified by PCR, linked to the *β-glucuronidase* (GUS) reporter gene, and transformed into *A. thaliana* in a testing vector of p1391Z by [39,40]. The ligation mixture was electroporated into *E. coli* DH5 (α) and was confirmed by PCR and subsequently into *A. tumefaciens* strain (EHA101) [39]. *Os*PPOGUS were then studied against drought and heavy metal treatments.

### 2.2. Plant Growth

Transgenic T2 (Homozygous) *A. thaliana* seeds were sterilized with 70% ethanol and were cultivated on Murashige and Skoog (MS) medium at half strength (1/2) [41]. Seedlings were grown in a growth chamber at 25 °C, 60% humidity, and a 16:8 light cycle for 25 days, and plants were used for stresses of drought and metals (Ni and Cu) applications (Figures 1–3).

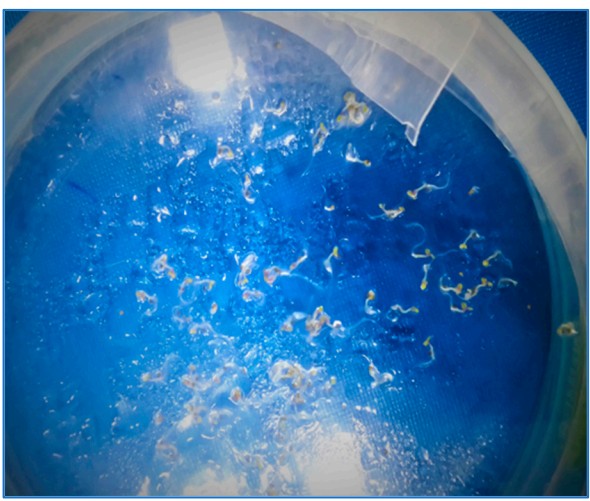

**Figure 1.** Small 4-day-old seedlings of T2 transgenic *A. thaliana*, grown on MS media.

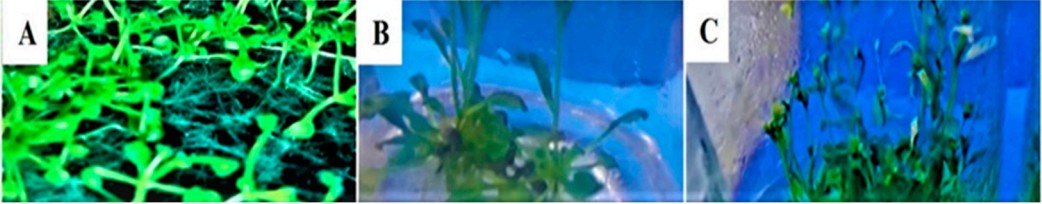

**Figure 2.** Seedlings of T2 transgenic *A. thaliana*, (**A**) 11-day-old, (**B**) 20-day-old, and (**C**) 24-day-old seedlings, were respectively grown on MS media.

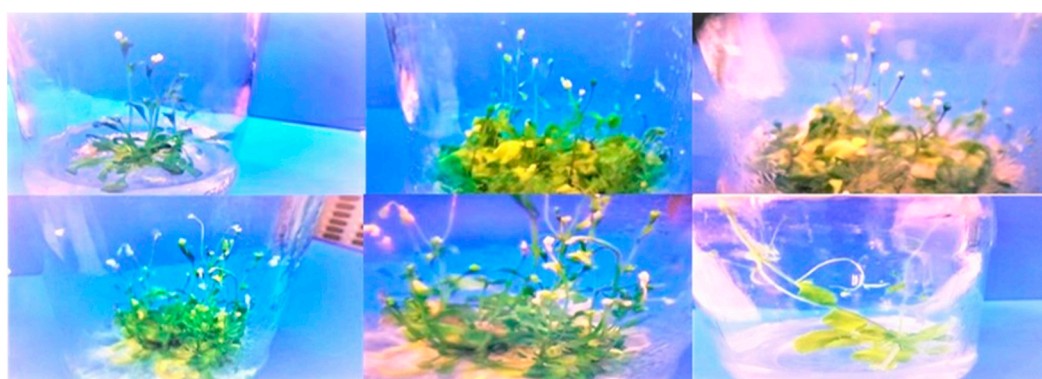

**Figure 3.** Flowering in multiple jars of T2 transgenic *A. thaliana* grown on MS media.

### 2.3. Drought

In this case, 25-day-old transgenic T2 plants were treated with drought stress using polyethylene glycol gel (PEG-4000). In each plate, 36 mL of MS medium was solidified with 0.8 percent agar, overlaid by 36 mL MS broth media containing 0, 5, 20, and 30% of PEG. The dissolved PEG yielded −0.05, −0.09, −0.58, and −1.8 megapascals (MPa) water potential. T2 plants were grown on these drought stress media for 6, 12, and 24 h. For

drought control, the same T2 transgenic plants were placed on media overlaid by liquid MS without PEG. Control samples remained on medium for 6, 12, and 24 h (Figure 4).

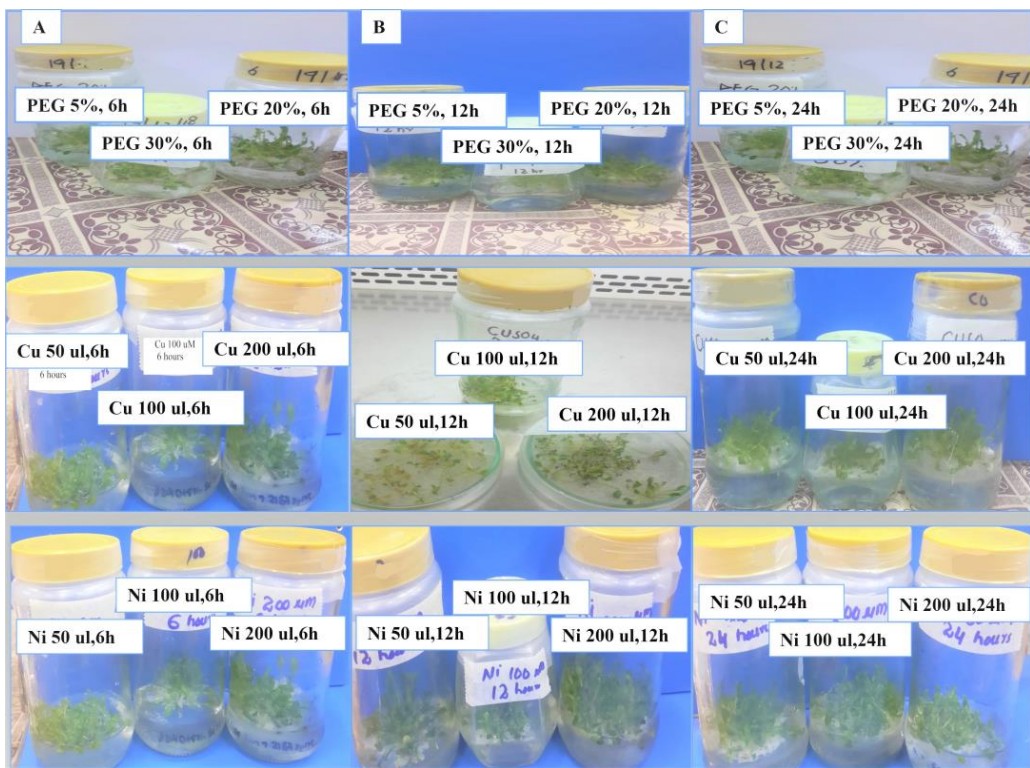

**Figure 4.** Polyethylene glycol, Cu, and Ni stress treatment at different concentrations (50 μM, 100 μM, and 200 μM) for 6 (**A**), 12 (**B**), and 24 (**C**) h, respectively, on T2 transgenic *A. thaliana* seedlings.

### 2.4. Heavy Metals
Cu and Ni Treatment

For Cu treatment, plants were sprayed with different concentrations (50 μM, 100 μM, and 200 μM) of copper and then analyzed after 6, 12, and 24 h. Similarly, (Ni) was applied to the second batch of transgenic *A. thaliana* by spraying with various concentrations of nickel (50 μM, 100 μM, and 200 μM) and then analyzed after 6, 12, and 24 h, respectively, shown in (Figure 4). Experimentation was repeated in triplicates.

### 2.5. GUS Staining

Gus staining is composed of 50 mM sodium dihydrogen phosphate, 10 mM disodium ethyl dimethyl tetra acetic acid (EDTA), and 0.01 percent Triton X 100 was used to make the GUS buffer (PH 7.0). To make an X-Gluc solution (0.1 M), 10 mg of X-Gluc was dissolved in 0.1 mL of DMSO. For GUS staining procedures, 1 mL of GUS buffer and 5 μL of 0.1 M X-Gluc solution were used. The plants were subjected to stress treatments (drought and heavy metals) before being submerged in GUS solution and incubated at 37 °C for the night. After the plants were discolored with methanol to remove any remaining chlorophyll, the relative GUS expression for each stress treatment was assessed based on the degree of GUS staining.

### 2.6. Enzymatic Assay

Currently, three activities were performed in each treatment, e.g., catalase (CAT), ascorbate peroxidase (APX), and peroxidase (POD).

### 2.6.1. Extract Preparations

The leaf tissue of plants was milled to a fine powder in liquid nitrogen with a tissue grinder in an extraction buffer [42]. Fresh tissues of the plants were homogenized in 4 mL of extraction buffer (Table 1). The homogenized mixture was centrifuged at 15,000 rpm for 22 min at 4 °C two times. The filtrate was used as an enzyme extract.

**Table 1.** Chemicals for extraction buffers.

| Chemical Name | Concentration | Volume |
|---|---|---|
| Potassium phosphate buffer | 25 mM (pH 7.8) | 1.4 mL |
| EDTA 100 mM | 0.4 mM | 16 μL |
| Ascorbic Acid 200 mM | 1 mM | 20 μL |
| Polyvinyl pyrmildone | 2% | 0.08 gm |
| Water | | 2.5 mL |
| Total | | 4 mL |

### 2.6.2. Catalase (CAT)

The activity of CAT was measured according to the [43] method. The absorbance reading was taken every 20 s at 240 nm at a minute interval to observe the trend of activity. Buffer solution pH 7.0 was used as a blank. The reaction mixture was used as a chemical constituent for the assay.

### 2.6.3. Ascorbate Peroxidase

The activity of APX was measured according to [44]. Absorbance readings were taken every 20 s at 290 nm at 2 min intervals. Buffer solution pH 7.0 was used as a blank. Chemicals for APX are prementioned in (Table 2).

**Table 2.** Chemicals for ascorbate peroxidase assay.

| Chemical Name | Concentration | Volume |
|---|---|---|
| Potassium phosphate buffer | 25 mM (pH 7.0) | 357 μL |
| Ascorbic acid (50 mM) | 0.25 mM | 6 μL |
| EDTA (100 mM) | 0.1 mM | 1.5 μL |
| $H_2O_2$ (100 mM) | 10 mM | 110 μL |
| Enzyme extract | 0.05 mL | 52 μL |
| Water | | 487 μL |
| Total | | 1 mL |

### 2.6.4. The Reaction Mixture for Peroxidase

An enzyme extract of 50 μL was mixed in the reaction mixture. Solutions of $H_2SO_4$ with a concentration of 5 N were prepared by adding 138.8 mL in 1 L of water. The absorbance value was measured at 485 nm (Table 3).

**Table 3.** Chemicals for peroxidase assay.

| Chemical Name | Concentration | Volume |
|---|---|---|
| Sodium phosphate buffer | 100 mM (pH 7.8) | 1.5 mL |
| Poly phenylenediamine | 4% | 1 mL |
| $H_2O_2$ | 1% | 1 mL |
| Enzyme extract | | 50 μL |
| Water | | 450 μL |
| $H_2SO_4$ | 5 N | 1 mL |
| Total | | 5 mL |

### 2.7. RNA Isolation and cDNA Synthesis

After treatments, samples from both control and stressed transgenic *A. thaliana* seedlings were harvested. Next, seedlings were crushed in liquid nitrogen, after which the ground sample was used for the extraction of total RNA. The RNA extraction was carried out manually using the method suggested by Oñate-Sánchez and Vicente-Carbajosa [45]. For quality and quantity confirmation, the Nano-drop technique (Nano-drop 1000 spectrophotometer, Thermo Scientific, Waltham, MA, USA) was used. After quantification, cDNA was synthesized using 1 μg of total RNA (Supplementary File).

### 2.8. Expression Analysis of OsPPO Promoter via Real-Time PCR (RT-PCR)

In the present study, primers housekeeping (actin) and reporter gene (GUS) was used to perform RT-PCR. For RT-PCR, the required reagent was supplemented in PCR tubes to make the total volume of the reaction mixture 11.16 μL. In an RT-PCR reaction, an increase in the fluorescence of "SYBR green" was seen, which corresponded to an increase in the expression level of GUS and actin genes (Table 4). Actin was used as a standard control for Arabidopsis [46]. This whole scenario is displayed in Figure 5.

**Table 4.** Sequences of actin and GUS primers.

| Primer | Primer Sequence |
|---|---|
| GUS F: | 5′ CGGCAGAGAAGGTACTGGAA 3′ |
| GUS R: | 5′ ATATCCAGCCATGCACACTG 3′ |
| Actintac F: | 5′ GATGAAGATACTCACAGAAAGA 3′ |
| Actintac R: | 5′ GTGGTTTCATGAATGCCAGCA 3′ |

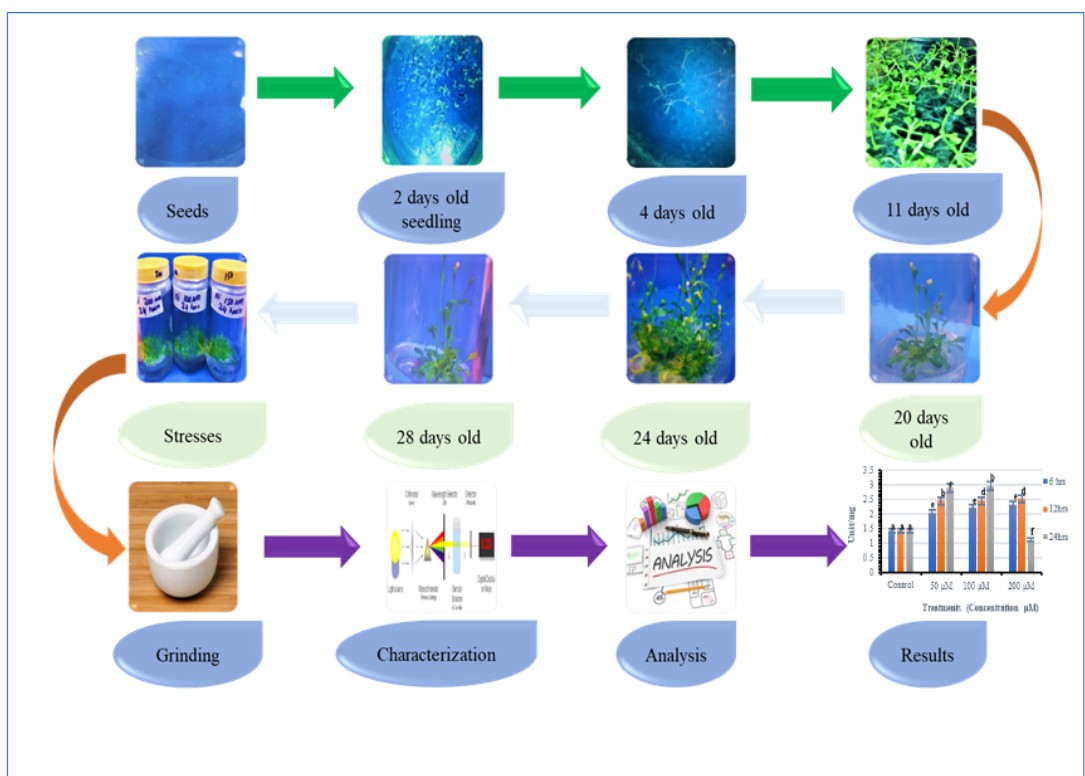

**Figure 5.** Detailed schematic summary of plant abiotic stresses and gene expression profiling in *A. thaliana*.

### 2.9. Statistical Analysis

The statistical analysis was carried out using the Statistics software version 10 (StatSoft, Inc., 1999, Analytical Software 2105 Miller Landing Rd, Tallahassee, FL 32312, USA). One-

way ANOVA was performed on the data, with various tests used to determine significant differences between means ($p \leq 0.05$) (Table 5).

**Table 5.** ANOVA table of *Os*PPOGUS mediated *A. thaliana* against abiotic stress.

| Variable | Minimum | Maximum | Mean | Std. Deviation | Mean Squares of Error | Significance |
|----------|---------|---------|------|----------------|----------------------|--------------|
| Ni | 0.100 | 7.780 | 1.628 | 2.247 | 4.905 | 0.46129 |
| Cu | 0.100 | 15.030 | 3.323 | 4.285 | 6.502 | 0.03743 |
| PEG | 0.030 | 1.900 | 0.279 | 0.520 | 0.255 | 0.43417 |

## 3. Results

### 3.1. GUS Staining

#### 3.1.1. Expression of *Os*PPOGUS in Response to PEG

Drought stress was applied to T2 plants while keeping control of T2 plants on basic MS media. Pictures with stains on them showed the relative GUS expression. In control plants, there was no sign of GUS activity. For 5% PEG, the *Os*PPO promoter's GUS activity first appeared, and it grew as the amount of drought stress increased. At 20%, it increases, becomes stronger, and stays that way until 30% PEG (Figure 6). Higher PEG concentrations resulted in an increasing pattern for *Os*PPOGUS activity.

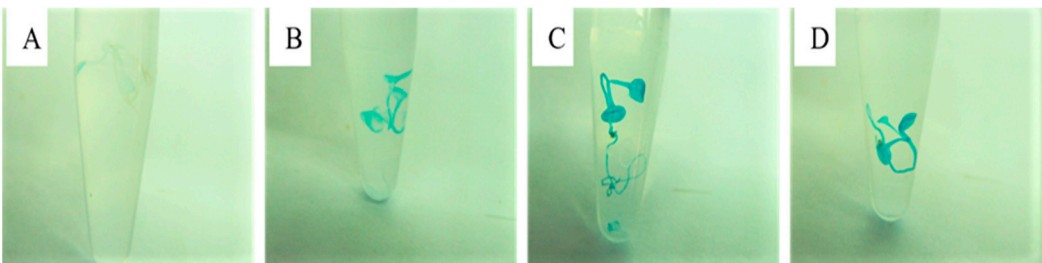

**Figure 6.** In response to various PEG concentrations, (**A**) control, (**B**) 5%, (**C**) 20%, and (**D**) 30% on MS for 24 h, *Os*PPOGUS expression was found in transgenic *A. thaliana* T2 plants.

#### 3.1.2. Expression of *Os*PPOGUS in Response to Heavy Metal Stress

In response to stress from heavy metals (Cu and Ni), PPO displayed essentially the same pattern of induction as was seen in response to drought stress. With higher heavy metal concentrations, GUS expression and intensity increase. It was noted in (Figures 7 and 8) that the control plant showed no pattern of induction. It was observed that the GUS expression increased up to 50 μM and 100 μM of Cu and Ni, then remained constant at 200 M. Therefore, it can be concluded that rising heavy metal concentrations cause the PPO promoter to respond.

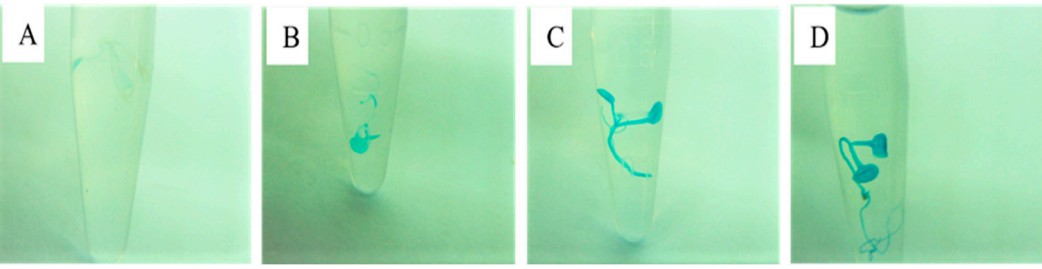

**Figure 7.** In response to various Cu concentrations, (**A**) control, (**B**) 50 μM, (**C**) 100 μM, and (**D**) 200 μM on MS for 24 h, *Os*PPOGUS expression was found in transgenic *A. thaliana* T2 plants.

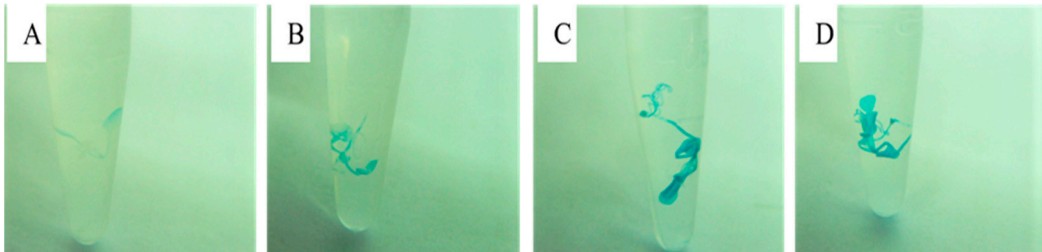

**Figure 8.** In response to various Ni concentrations, (**A**) control, (**B**) 50 µM, (**C**)100 µM, and (**D**) 200 µM on MS for 24 h, *Os*PPOGUS expression was found in transgenic *A. thaliana* T2 plants.

### 3.2. OsPPOGUS Response to Drought Treatment

Drought (PEG 4000) stress was applied to transgenic Arabidopsis T2 lines. Based on RT-PCR results, T2 lines of transgenic Arabidopsis showed an expression level of *Os*PPOGUS in response to drought treatment. Upon 5% PEG stress, maximum fold induction of 0.09 was observed after 24 h, while after 6 and 12 h, it showed lower fold induction (0.07 and 0.03, respectively). The maximum fold induction (0.43) was observed after 12 h for 20% PEG after 12 h of treatment, and lower induction (0.19- and 0.14-fold) was revealed after 24 and 6 h drought stress. Similarly, for 30% PEG, maximum fold induction (1.9) was seen after 12 h, while reduced fold induction (0.1) was observed after 24 and 6 h of drought stress, respectively. Overall, various concentrations (5%, 20%, and 30%) induced PPO expression after 6, 12, and 24 h are shown in Figure 9.

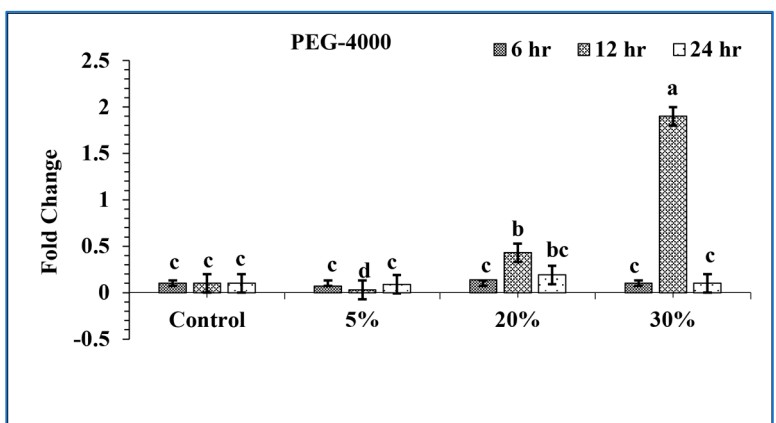

**Figure 9.** RT-PCR study of *Os*PPOGUS activity with different concentrations of PEG. qRT-PCR was used to detect GUS gene transcript levels in 25-day-old, transformed Arabidopsis plants growing on MS media by spraying various concentrations of PEG (5%, 20%, and 30%) solutions. The data revealed mean ± standard deviation (SD). Various letters (a, b, c, and d) specify significantly different results from ANOVA post hoc Tukey-adjusted comparisons of three independent tests ($p \leq 0.05$, $n = 3$).

### 3.3. Induction of OsPPOGUS in Response to Ni Treatment

Based on RT-PCR results, it was observed that the *Os*PPOGUS expression level was enhanced in transgenic Arabidopsis in response to heavy metal (Ni) stress. Maximum fold induction (1.197) with 50 µM Ni stress was found after 24 h, while after 12 and 6 h, it showed lower fold induction (1.14 and 0.19), respectively. Similarly, for 200 µM Ni treatment, maximum fold induction was observed (4.33) after 12 h, and a lower fold induction (1.12 and 1.17) was noted in the time intervals of 24 h and 6 h, respectively. Moreover, the maximum fold induction (7.78) for 100 µM Ni stress treatment was found again after 24 h and lower expression was found after 12 h (1.17) and 6 h (1.14). Overall, *Os*PPO responded to various concentrations (50 µM, 100 µM, and 200 µM) of Ni after their

respective time of 6, 12, and 24 h, collectively displayed in Figure 10. In general, it was observed that *Os*PPOGUS showed the best response to Ni treatment.

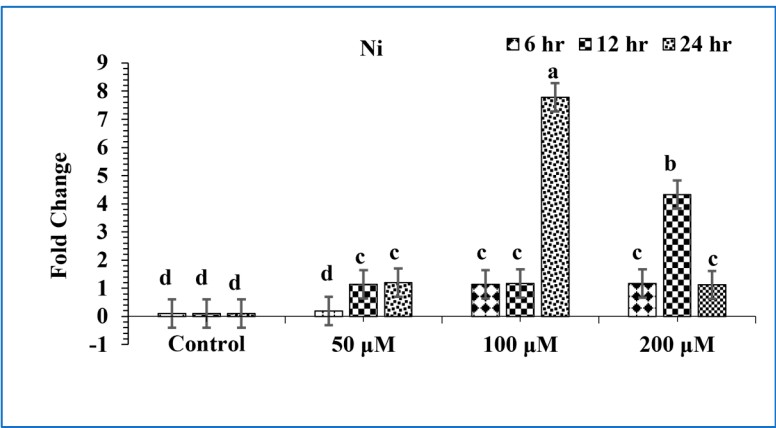

**Figure 10.** RT-PCR study of *Os*PPOGUS activity with different concentrations of Ni. qRT-PCR was used to detect GUS gene transcript levels in 25-day-old, transformed Arabidopsis plants growing on MS media by spraying various concentrations of Ni (50, 100, 200 µM) solutions. The data revealed mean ± standard deviation (SD). Different letters (a, b, c, and d) specify significantly different results from ANOVA post hoc Tukey-adjusted comparisons of three independent tests ($p \leq 0.05$, $n = 3$).

### 3.4. OsPPOGUS Response to Cu Treatment

Copper stress was applied to transgenic Arabidopsis T2 lines. The expression level of reporter gene GUS was analyzed via RT-PCR. Based on RT-PCR results, T2 lines of transgenic Arabidopsis showed an enhanced expression level of *Os*PPOGUS in response to Cu treatment. For 50 µM Cu, maximum fold induction of 6.61 was observed after 24 h, while after 12 and 6 h, it showed lower fold induction (0.96 and 1.11, respectively). The maximum fold induction (15.03) was observed after 24 h of 100 µM Cu treatment, and lower induction (3.38- and 4.42-fold) was observed after 12 and 6 h of Cu stress, respectively. Similarly, for 200 µM Cu, the maximum fold induction (5.43) was seen after 24 h, while reduced fold induction (1.04 and 0.49) was observed after 12 and 6 h of Cu stress, correspondingly. Overall, results elucidated *Os*PPO response to various concentrations (50, 100, and 200 µM) of Cu after 6, 12, and 24 h are shown collectively in Figure 11.

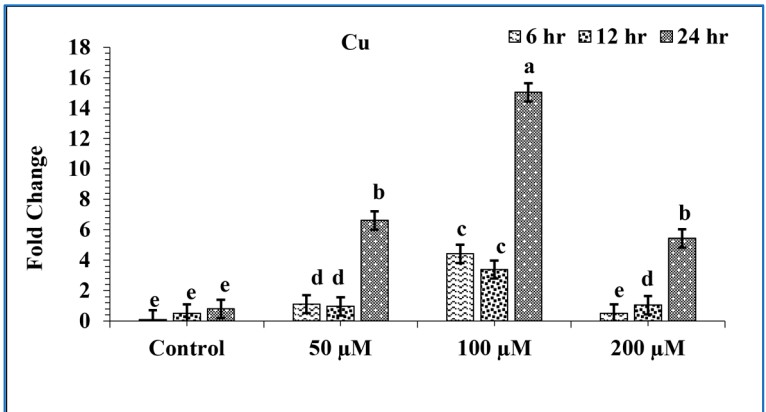

**Figure 11.** The activity of *Os*PPOGUS through RT-PCR study with different Cu concentrations. Quantitative RT-PCR was used to detect GUS gene transcript levels in 25-day-old, transformed Arabidopsis T2 lines plants growing on MS media by spraying Cu (50, 100, 200 µM) solutions. The data revealed mean ± standard error (SE). Various letters (a, b, c, d, and e) specify significantly different results from ANOVA post hoc Tukey-adjusted comparisons of three independent tests ($p \leq 0.05$, $n = 3$).

### 3.5. Quantification of ROS and Antioxidant Assay in A. thaliana

Antioxidant enzymes such as SOD, POD, and APX all protect Arabidopsis plants from environmental stress, including drought and heavy metal stress. Specifically, (superoxide dismutase) helps the plant reduce the impact of oxidative stress caused by reactive oxygen species, POD (peroxidase) helps remove hydrogen peroxide, and APX (ascorbate peroxidase) helps reduce levels of hydroperoxides. In the current study, three activities were performed in each treatment, e.g., catalase (CAT), ascorbate peroxidase (APX), and peroxidase (POD) against drought and heavy metal stress. All three enzymes have been found to be upregulated in Arabidopsis plants under drought and heavy metal stress, indicating that they play an important role in protecting the plant from stress (Figure 12A–C, respectively).

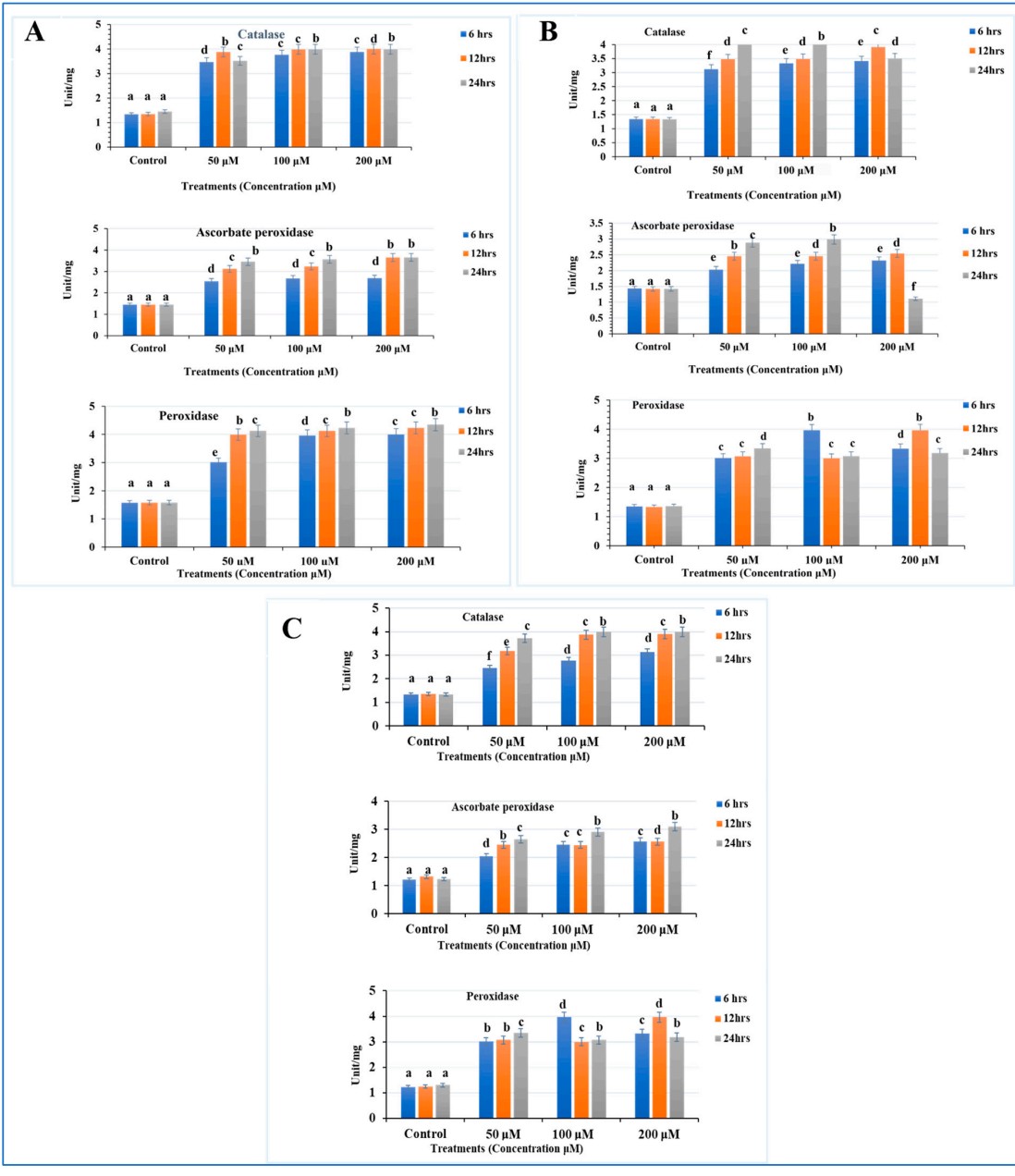

**Figure 12.** Estimation of antioxidant enzyme assay, e.g., CAT, APX, and POD in transgenic Arabidopsis stressed with different concentrations of PEG (**A**), Ni (**B**), and Cu (**C**) stresses. The data are presented as the mean ± SD. Various letters (a, b, c, d, e, and f) specify significantly different results from ANOVA post hoc Tukey-adjusted comparisons of three independent tests ($p < 0.05$, $n = 3$).

## 4. Discussion

Abiotic stressors are widely acknowledged to be key limiting factors affecting plant growth and development worldwide. Numerous stresses have a negative impact on plant development and agricultural production. As a result, it is important to investigate how plants adapt to various compressions [47,48]. The phenylpropanoid route is most likely the best-researched secondary metabolic pathway in plants. As ubiquitous copper metalloenzymes, PPOs have a significant role in plant growth, development, and stress tolerance [49,50]. This enzyme might function by either direct toxicity of quinones created by its catalytic activity or by cross-linking them with proteins, producing physical barriers to external pressure. The increase in plant resistance is connected to the numerous roles of polyphenols in plants, which are collections of compounds that protect plants from excessive light, such as UV (flavonoids) and visible light (anthocyanins). The role of PPO in plants, response to heavy metals, or abiotic stresses has been demonstrated.

Currently, the level of *Os*PPO fused with GUS against drought and heavy metal (Cu and Ni) stress in transgenic *A. thaliana* was investigated. In the case of drought, maximum fold induction (1.9) was observed after 12 h, and minimum (0.07) fold induction was noted after 12 h. In the case of heavy metals, both (Cu and Ni) displayed the maximum fold induction (15.03-fold) and (7.78-fold), respectively, for 100 μM after 24 h. This expression data revealed that PPO is highly functional under these stresses. A study was conducted to examine the effect of triadimefon (TDM) on drought stress in sunflower (*Helianthus annuus*) plants, and it was found that PPO activity increased when compared to the control [51]. The action of the PPO effect by water stress (drought) has earlier been evidenced in tomatoes [52]. In addition to the foregoing findings, it was shown that *R. serbica* dried leaves induced some fold greater PPO induction when plants were subjected to near-complete drought conditions [53]. Furthermore, water stress was observed to increase PPO induction in coconut and wheat [54]. Ref. [55] Correlating these findings with *Os*PPO promoter drought responsiveness is similar to GUS expression of potato PPO in transgenic tomatoes in response to drought, with the maximum extent in adult leaves and abscission zones of the plant to manage drought.

Apart from drought, PPO is being affected by applying heavy metal stress. This observation supports preceding findings of *Annona muricata* leaves revealed higher induction of PPO against Cu [56]. Present results match seamlessly with that of [57], who confirmed the antioxidative properties of nickel in soybean cell cultures under heavy metal stress. The study in [58] shows the action efficiency and bid of Hydrilla verticillata and may provide a reference for heavy metal phytoremediation and boost the activity of PPO. Abiotic stressors also stimulate cell signaling, which upregulates the phenylpropanoid pathway transcription [59]. The capability of various polyphenol classes to shield the plant from too much light, such as UV (flavonoids) and visible light, as well as their ability to scavenge ROS, are two of the multiple roles of polyphenols in plants that are associated with an increase in plant resistance (anthocyanins). Under abiotic stress, polyphenols may also perform important ecological roles, such as acting as signaling molecules for other plants.

## 5. Conclusions

Currently, the higher expression of *Os*PPOGUS in transgenic Arabidopsis in response to drought and heavy metal (Cu and Ni) stress shows the suitability of the plant species. The expression data of *Os*PPOGUS revealed that PPO has the potential to be expressed under drought and heavy metal (Cu and Ni) stress. PPO enhances the production of phenolic compounds, which trigger ROS in plants and consequently improve plant tolerance to abiotic stress. In the future, a holistic strategy that considers many management choices for dealing with heat and drought stress at the same time might be a win–win situation.

**Supplementary Materials:** The following supporting information can be downloaded at: https://www.mdpi.com/article/10.3390/su151712783/s1, Figure S1: RNA confirmation via gel elec-

trophoresis; Table S1: Reaction components for isolation of RNA; Table S2: Constituents of the reaction mixture for RT-PCR.

**Author Contributions:** Z.U. designed and conceived the study. Z.U. and J.I. completed the experiments. B.A.A., W.A., S.K. and I.A. analyzed the data. Z.U. and W.C. performed the visualizations and statistical data analysis. Z.U. and J.I. wrote the original draft. Z.U., J.I., B.A.A., W.A., I.A. and M.A.E.-S. reviewed and edited the manuscript. M.A.E.-S. also provided funds. T.M. provided the resources. Z.U. and W.C. made valuable revisions and edited the manuscript. All authors have read and agreed to the published version of the manuscript.

**Funding:** This research was funded by King Saud University, Riyadh, Saudi Arabia, grant number RSP2023R182.

**Institutional Review Board Statement:** Not applicable.

**Informed Consent Statement:** Not applicable.

**Data Availability Statement:** All the raw data of this research can be obtained from the corresponding authors upon reasonable request.

**Acknowledgments:** The authors would like to extend their sincere appreciation to the Researchers Support Project Number (RSP2023R182), King Saud University, Riyadh, Saudi Arabia.

**Conflicts of Interest:** The authors declare no conflict of interest.

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
